# Removal of reinforcement improves instrumental performance in humans by decreasing a general action bias rather than unmasking learnt associations

**Hannah Kurtenbach**[ID]\*, **Eduard Ort**[ID], **Monja Isabel Froböse**[ID], **Gerhard Jocham**

Biological Psychology of Decision Making, Institute of Experimental Psychology, Heinrich Heine University Düsseldorf, Germany

\* hannah.kurtenbach@hhu.de

**Data Availability Statement:** All data and codes are made available on OSF (https://osf.io/nxhd5/).

## Abstract

Performance during instrumental learning is commonly believed to reflect the knowledge that has been acquired up to that point. However, recent work in rodents found that instrumental performance was enhanced during periods when reinforcement was withheld, relative to periods when reinforcement was provided. This suggests that reinforcement may mask acquired knowledge and lead to impaired performance. In the present study, we investigated whether such a beneficial effect of removing reinforcement translates to humans. Specifically, we tested whether performance during learning was improved during non-reinforced relative to reinforced task periods using signal detection theory and a computational modelling approach. To this end, 60 healthy volunteers performed a novel visual go/no-go learning task with deterministic reinforcement. To probe acquired knowledge in the absence of reinforcement, we interspersed blocks without feedback. In these non-reinforced task blocks, we found an increased $d'$, indicative of enhanced instrumental performance. However, computational modelling showed that this improvement in performance was not due to an increased sensitivity of decision making to learnt values, but to a more cautious mode of responding, as evidenced by a reduction of a general response bias. Together with an initial tendency to act, this is sufficient to drive differential changes in hit and false alarm rates that jointly lead to an increased $d'$. To conclude, the improved instrumental performance in the absence of reinforcement observed in studies using asymmetrically reinforced go/no-go tasks may reflect a change in response bias rather than unmasking latent knowledge.

## Author summary

It appears plausible that we can only learn and improve if we are told what is right and wrong. But what if feedback overshadows our actual expertise? In many situations, people learn from immediate feedback on their choices, while the same choices are also used as a measure of their knowledge. This inevitably confounds learning and the read-out of learnt

**Funding:** The work was supported by a European Research Council grant (ERC-CoG 771432) to GJ. The funders had no role in study design, data collection and analysis, decision to publish, or preparation of the manuscript.

**Competing interests:** The authors have declared that no competing interests exist.

associations. Recently, it was suggested that rodents express their true knowledge of a task during periods when they are *not* rewarded or punished during learning. During these periods, animals displayed improved performance. We found a similar improvement of performance in the absence of feedback in human volunteers. Using a combination of computational modelling and a learning task in which humans' performance was tested with and without feedback, we found that participants adjusted their response strategy. When feedback was not available, participants displayed a reduced propensity to act. Together with an asymmetric availability of information in the learning environment, this shift to a more cautious response mode was sufficient to yield improved performance. In contrast to the rodent study, our results do not suggest that feedback masks acquired knowledge. Instead, it supports a different mode of responding.

## Introduction

In everyday life it is crucial to learn whether an action leads to reward or punishment. This adaptive behaviour has been extensively investigated in animal and human experiments and formally captured using reinforcement learning models [1–4]. In these models, the expected value of an action is updated using prediction errors, which reflect the discrepancy between obtained and expected rewards, in order to optimize future choices. Most learning tasks measure task performance while feedback is provided, which inevitably confounds learning with instrumental performance. To decouple learning and instrumental performance, some studies feature a learning phase and a later probe phase in which knowledge is tested in the absence of feedback. These studies show that different neural mechanisms underlie learning and expression of knowledge [5–9]. However, in these studies, acquired knowledge is usually tested *after* the learning performances has reached a plateau. In contrast, little is known about what happens when knowledge is tested without reinforcement during the learning process *prior to* participants reaching asymptotic performance.

During perceptual learning tasks, absence of feedback resulted in impaired [10], or unchanged [11] performance in humans. These results contrast with a recent rodent study in the domain of associative learning: Omitting feedback during early learning improved performance. Notably, performance deteriorated again when reinforcement was reintroduced, suggesting that reinforcement masked the underlying knowledge acquired by the animals [12].

The present study investigates whether this finding, that has been observed in rodents, extends to human learning. Specifically, we asked whether healthy volunteers' performance benefits similarly from omitting reinforcement during instrumental learning. To this end, closely following Kuchibhotla and colleagues [12], we adopted a go/no-go task that required participants to learn, by trial and error, to respond to go stimuli to obtain reward and to withhold responding to no-go stimuli to avoid punishment (monetary wins and losses, respectively). Crucially, reinforced trials were interleaved with multiple blocks in which participants were instructed to continue responding as previously, but no reinforcement was delivered (probe blocks). Similar to the pattern observed in rodents [12], we found that performance, as quantified by the sensitivity index *d'*, was improved in probe blocks, relative to reinforced blocks. However, computational modelling revealed that this pattern did not result from an increased sensitivity to acquired values. Instead, the behavioural pattern in the present paradigm could be completely explained by a mere reduction of an overall propensity to respond. Together with an initial tendency to act (as reflected in a positive initialization of value estimates), this change in overall response bias is sufficient to cause asymmetric changes to hit

and false alarm rates that jointly lead to an increased sensitivity index $d'$. Altogether, these results support the notion that omission of reinforcement may improve instrumental performance, however, rather than unmasking latent associative knowledge, this is due to a change in the overall propensity to act.

## Results

### Task

Based on recent findings in rodents [12], we hypothesized that the performance of humans in an instrumental learning task increases during non-reinforced compared to reinforced periods. Therefore, we designed a visual go/no-go reinforcement learning task (Fig 1A) consisting of reinforced trials which were interleaved with five probe blocks of non-reinforced trials. We used twelve greebles as stimuli [13]. Half of them were randomly assigned as go options, while the other half was assigned as no-go options. On each trial, one of the twelve stimuli was presented and participants had to learn, from trial and error, to perform a button press for go stimuli and to withhold responding for no-go stimuli. We used a rather high number of stimuli to be learnt by participants in order to evoke slow, incremental learning (Fig 1B). Participants obtained reward (monetary gain) for responding to go stimuli and punishment (monetary loss) for responding to no-go stimuli. Withholding a response resulted in no feedback (and neither monetary gain nor loss). This asymmetric reinforcement schedule follows the design used by Kuchibhotla and colleagues [12] and other work in rodents [14,15]. During probe trials, participants were instructed to continue choosing as they would do during reinforced trials, while reinforcement was temporarily omitted. We performed two experiments: an original study ($N = 30$) and a replication in an independent sample ($N = 30$). The main results were similar across the two studies; therefore, here we report the results of the pooled sample (see Section 1 in S1 Appendix for a separate presentation).

### Analysis approaches: Signal detection theory and computational modelling

To assess effects of the removal of reinforcement on task performance, we present two approaches. First, we aimed to replicate the results of Kuchibhotla and colleagues' work [12] based on signal detection theory (SDT). To this end, we computed the sensitivity index $d'$, representing the difference in the relative frequencies of hits and false alarms (button presses to go and no-go stimuli, respectively) [16]. To compute measures from SDT, it is necessary to consider windows of several trials. Importantly, this approach can introduce artefacts in learning paradigms, due to its insensitivity to the general rising trend in performance. Specifically, during learning mean performance on earlier trials is intrinsically lower than mean performance on later trials, irrespective of any manipulation, such as feedback removal. Consequently, $d'$ implicitly disadvantages earlier trials compared to later ones, unless performance has reached stable levels. For this reason, we next present a computational modelling approach, which averts this issue by providing trial-by-trial estimates. Conclusions are primarily based on this second approach, presented in the paragraph "Computational modelling reveals a shift in action bias".

### Sensitivity index $d'$ is increased when reinforcement is omitted

The SDT measure $d'$ indicated a gradual increase in participants' performance (Fig 1B), confirming successful acquisition of the correct associations over time. This increase is mostly driven by no-go trials: While the initial go-response probability for no-go trials is high, participants learn to withhold responding over the course of the experiment (Fig 1C). To statistically

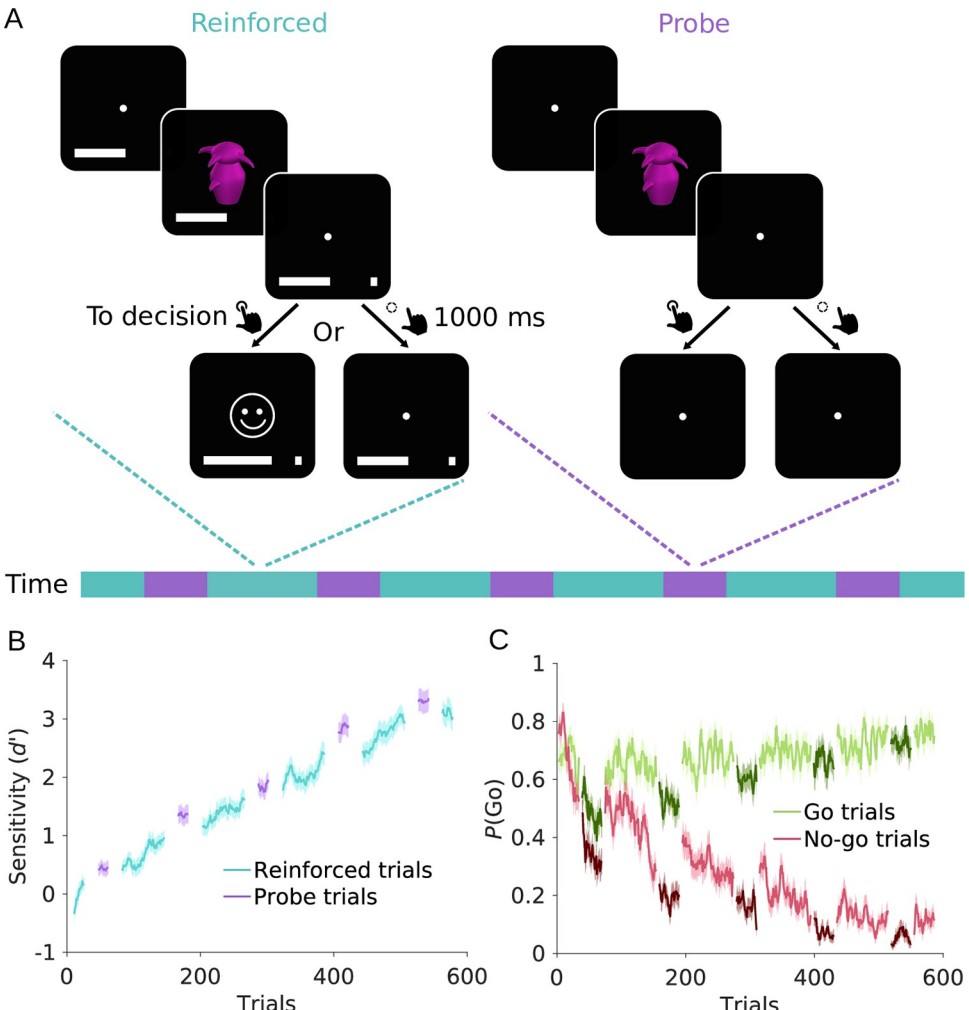

**Fig 1. Task structure and participants' behaviour.** (A) Schematic of the go/no-go learning task. On each trial, a fixation cross was presented for 1000–1600 ms. Then, participants were presented with one stimulus for 500 ms and had 1000 ms to decide whether to perform a go (button press) or no-go (no button press) response. Blocks of reinforced trials alternated with probe blocks (illustrated in the timeline). On reinforced trials (cyan), a go response resulted in reward or punishment (monetary win or loss, indicated by a smiley or frowny, respectively), depending on whether the stimulus was a go or no-go stimulus. No-go responses resulted in no feedback, and in neither reward nor punishment. A progress bar at the bottom of the screen displayed cumulative reward (rewards increased the bar, punishments shrank it). On probe block trials (purple), participants were required to respond as during reinforced blocks, but no feedback following responses was provided. (B) Sensitivity index *d'*, separately for reinforced (cyan) and probe trials (purple). (C) Time course of go-response probabilities, *P*(Go), for go trials (green) and no-go trials (red). Darker shades of green and red indicate probe trials. Solid lines in B and C represent mean, shaded areas SEM across participants.

assess the change from reinforced to probe blocks, we compared performance of the 36 trials in a probe block with performance in the 36 trials before each probe block (pre-probe trials). Across the entire experiment, *d'* was indeed higher for probe compared to pre-probe trials ($\Delta d'$ = 0.47 ± 0.53, mean ± SEM, $t_{59}$ = 6.94, $p < .001$, Cohen's *d* = 0.90, Fig 2A). This increase in *d'* was driven by a more pronounced reduction of false alarm rate (*FAR*) than hit rate (*HR*; $\Delta FAR$–$\Delta HR$ = -0.07 ± 0.08, $t_{59}$ = -6.42, $p < .001$, Cohen's *d* = -0.83). However, both measures decreased significantly in probe compared to pre-probe trials ($\Delta HR$ = -0.08 ± 0.06, $t_{59}$ = -9.67, $p < .001$, Cohen's *d* = -1.25; $\Delta FAR$ = -0.15 ± 0.08, $t_{59}$ = -15.12, $p < .001$, Cohen's *d* = -1.95, Fig

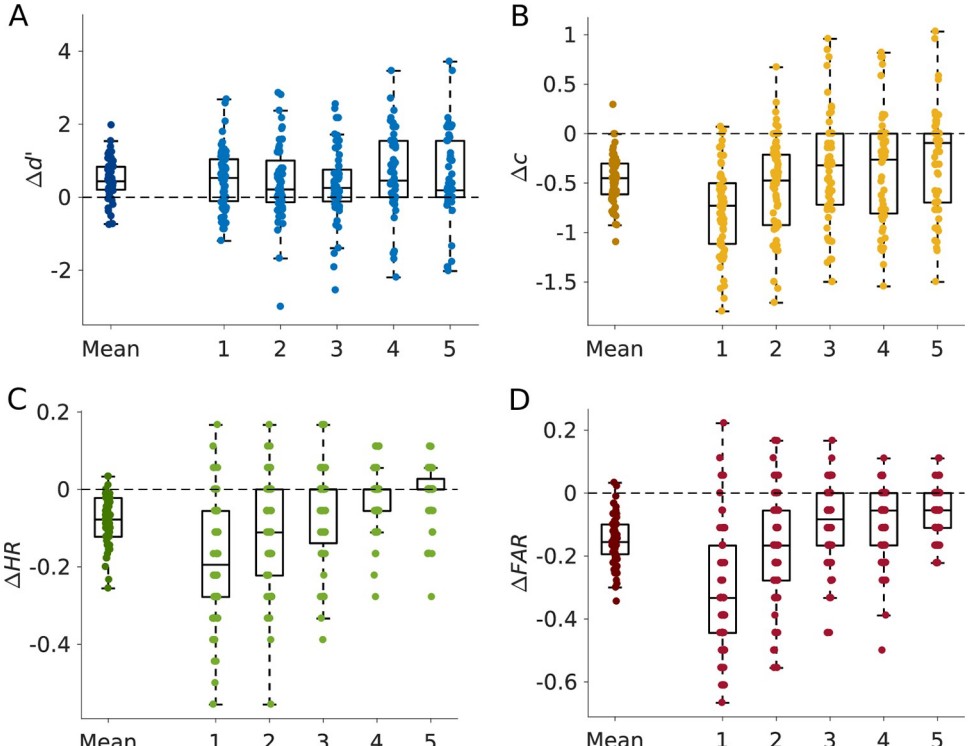

**Fig 2. Behavioural results, expressed as difference between probe trials and preceding reinforced trials.** Results are shown both for the mean across all five probe blocks (left) and separately for each probe block. Points reflect individual participants' behaviour. (A) The sensitivity index $d'$ increased in probe compared to reinforced trials. (B) The negative bias criterion $c$ decreased on probe blocks, indicating a reduced propensity to act on probe trials. (C), (D) Both hit rate (*HR*, C) and false alarm rate (*FAR*, D) decreased on probe blocks, but the decrease in *FAR* was more pronounced than the decrease in *HR*, which lead to the increase in $d'$ represented in (A).

2C and 2D), while in [12] only a decrease in false alarm rate during probe trials was reported. Once reinforcement was reinstated, $d'$ significantly decreased again ($\Delta d' = 0.15 \pm 0.47$, $t_{59} = 2.40$, $p = .020$, Cohen's $d = 0.31$, see Fig G in S1 Appendix). Note that this effect was only evident in Experiment 2 when analysing the two experiments separately (see Section 1 in S1 Appendix). Again, the decrease in $d'$ was driven by a significant increase in both hit and false alarm rate ($\Delta HR = -0.09 \pm 0.07$, $t_{59} = -7.92$, $p < .001$, Cohen's $d = -1.02$; $\Delta FAR = -0.10 \pm 0.10$, $t_{59} = -9.80$, $p < .001$, Cohen's $d = -1.27$, see Fig G in S1 Appendix).

In rodents, the removal of feedback improved performance only during early learning [12]. Therefore, we hypothesized that the increase in $d'$ is strongest for early probe blocks. Contrary to this, there was no effect of time on the change in $d'$ over probe blocks ($F(4, 59) = 0.74$, $p = .568$, $\eta^2 = 0.35$, Fig 2A), despite time effects on hit and false alarm rates (hit rate: $F(4, 59) = 24.18$, $p < .001$, $\eta^2 = 0.01$; false alarm rate: $F(4, 59) = 32.32$, $p < .001$, $\eta^2 = 0.29$, Fig 2C and 2D). Post hoc tests confirmed a significant increase in $d'$ from pre-probe to probe trials for all five probe blocks (all $t_{59} \geq 2.74$, $p \leq .008$, see Table I in S1 Appendix). Thus, the increase in $d'$ was not specific to early learning. The comparison of probe trials with post-probe trials yielded similar results (see Table J in S1 Appendix).

In addition to the sensitivity index $d'$, we also quantified the change in response bias of participants between probe and reinforced trials using the bias criterion $c$ from SDT [16,17]. The bias criterion decreased from pre-probe to probe ($\Delta c = -0.47 \pm 0.25$, $t_{59} = -14.56$, $p < .001$, Cohen's $d = -1.88$, Fig 2B) and increased again in post-probe trials ($\Delta c = -0.42 \pm 0.32$, $t_{59} =$

-10.22, $p < .001$, Cohen's $d$ = -1.32, see Fig G in S1 Appendix), thus, go-responding was reduced during probe trials, but increased when reinforcement was re-introduced again. Furthermore, there was an effect of time on the change of bias criterion from pre-probe to probe trials ($F(4, 59) = 9.21$, $p < .001$, $\eta^2 = 0.14$, Fig 2B), indicating that the reduced go-responding in probe trials diminished over the experiment.

In summary, both hit and false alarm rates decreased during probe compared to reinforced blocks, leading to a reduced response bias $c$. However, the decrease in false alarm rates was more pronounced compared to the decrease in hit rates, which further resulted in an increased sensitivity index $d'$.

## Computational modelling reveals a shift in action bias

Due to the confound with SDT-based parameters in learning experiments described above, those results cannot be used here to distinguish between a real effect of reinforcement removal and an artefactually introduced effect. To overcome this issue, we used computational modelling. Unlike measures like $d'$ which require consideration of several trials, reinforcement learning models provide value estimates for all stimuli for each trial [18–20]. We used variants of $Q$-learning with a delta update rule and softmax action selection. Two key parameters are at the heart of these reinforcement learning models: a learning rate and a softmax choice temperature. The learning rate determines the extent to which prediction errors are used to update value estimates, hence, governing the speed of learning. The softmax choice temperature determines how sensitive choices are to acquired value: At higher temperatures, participants' choices are increasingly stochastic, and large values are required to select the correct choice with high probability, while at low softmax temperatures, values slightly greater/lower than zero are sufficient to reliably select the go/no-go action, respectively. While in multi-alternative decisions, the temperature governs the balance between exploration and exploitation, in our paradigm with a single option per trial and deterministic reinforcement, the temperature can be used as an index of choice sensitivity. Critically, since the temperature is fitted based on trial-wise value estimates, it is not subject to the issues $d'$ entails.

To test whether choice sensitivity in probe trials was indeed improved compared to reinforced trials or whether improved performance resulted from a non-specific change in action bias, we set up and compared four different models: a *baseline model*, a *temperature model*, a *bias model* and a *full model*. In all four models, learning rates $\alpha$ were set to a fixed value of 0.06 because learning rate and softmax temperature are strongly correlated for deterministic task structures (see methods for detailed reasoning) [21]. To rule out that the results are specific to this particular choice of learning rate, we re-ran all analyses across a wide range of learning rates and obtained an identical pattern of results (see Section 3.1. in S1 Appendix).

The baseline model included four free parameters: one single softmax temperature $\tau$ and a bias term $b$ for both block types, one initial value estimate $Q_0$ for each of the twelve stimuli, and a decay parameter $\theta$. On reinforced trials in which the go action was selected (and feedback received), values were updated using a delta update rule. On probe trials in which the go action was selected (and no feedback received), no changes were applied to value estimates. During both reinforced and probe trials without a go action, we assumed that values were subject to passive forgetting [22,23] via diffusion towards zero governed by the decay parameter $\theta$. In addition, the softmax choice rule contained a bias term $b$ that indicates participants' overall propensity to respond, independent of the option's current value. The temperature model was based on the baseline model, but it featured separate temperatures $\tau_R$ and $\tau_P$, for reinforced and probe trials, respectively. Similarly, the bias model was based on the baseline model, but now, instead of the temperature, we allowed the bias parameter $b$ to be different for reinforced

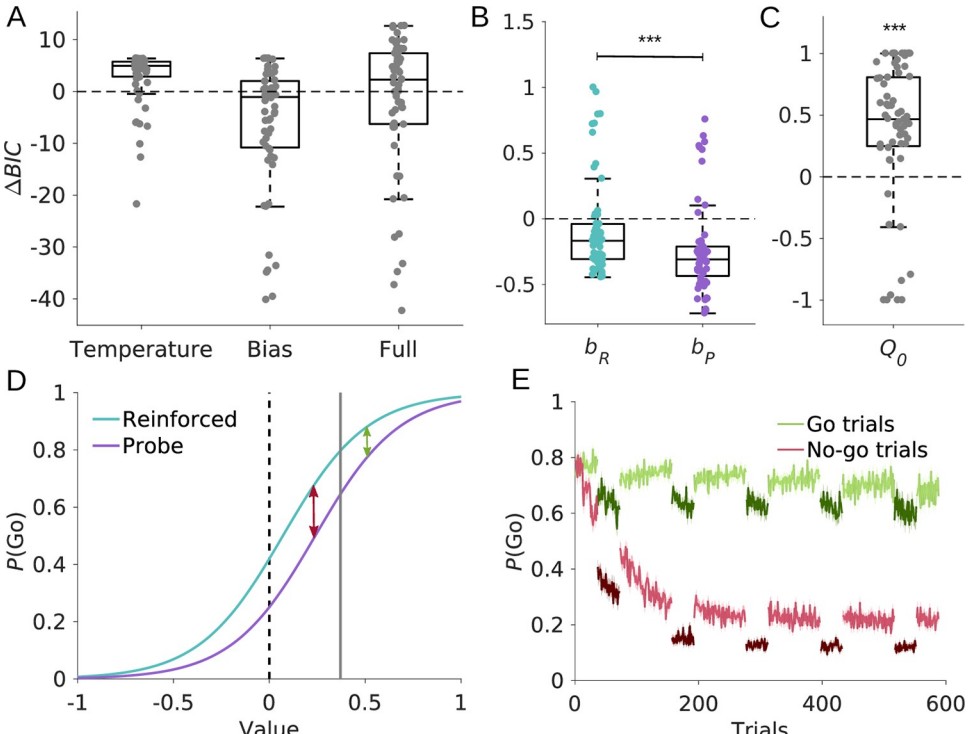

**Fig 3. Computational modelling results.** (A) Comparison of the Bayesian information criterion (*BIC*) relative to the baseline model. Negative *BIC* differences indicate a decrease in BIC relative to the baseline model and hence better fit. Conversely, a positive *BIC* difference indicates worse fit. The bias model provided the best fit. (B) The bias model contained two separate bias parameters, $b_R$ and $b_P$, for reinforced and probe blocks, respectively. The bias is reduced on probe compared to reinforced trials. (C) Initial estimates $Q_0$ of option values. On average, estimates were initialized with positive values. (D) Softmax choice probabilities to select an option as a function of its value. The sigmoids for reinforced and probe trials were generated using the mean fitted parameters. This figure illustrates how a reduction in response bias together with a positive value initialization resulted in the increase in *d'* observed in behaviour. Solid vertical grey line indicates average $Q_0$. As values of go stimuli were acquired (shifting rightwards from the vertical line), the difference in action probabilities between probe and reinforced trials became smaller (green arrow). Conversely, as values of no-go stimuli were acquired (shifting leftwards from the vertical line), the difference became more pronounced (red arrow), thus leading to a stronger reduction in false alarm rates. (E) Time course of simulated go-response probabilities. The probability $P$(Go) for go trials (green) and no-go trials (red) was simulated based on the bias model. Darker shades of green and red indicate probe trials. Solid lines represent mean, shaded areas SEM across simulations.

and probe trials ($b_R$ and $b_P$). Finally, the full model incorporated both separate temperature parameters $\tau_R$ and $\tau_P$ and separate bias parameters $b_R$ and $b_P$.

Contrary to our expectation, the temperature model performed the worst (*BIC* = 566.90 ± 137.25, median ± SEM, Fig 3A), followed by the full model (*BIC* = 563.48 ± 132.10), outperformed even by the baseline model (*BIC* = 561.56 ± 137.39). The best fitting model was the bias model (*BIC* = 557.86 ± 131.99). Closer analysis of the bias model showed that the response bias in reinforced trials, $b_R$, was higher compared to the bias in probe blocks, $b_P$ (*Δb* = -0.17 ± 0.12, $t_{59}$ = -10.92, $p < .001$, Cohen's *d* = -1.41, Fig 3B). Thus, participants had a reduced propensity to act during probe blocks.

One might argue that the differences in behaviour between reinforced and probe blocks are subtle, such that the improvement in model fit conferred by two separate temperatures in the full model did not survive punishment by the Bayesian information criterion. We therefore explored the full model and tested whether there were differences in either temperatures, $\tau_R$ and $\tau_P$, or bias parameters, $b_P$ and $b_R$, or both. Again, we found that the bias parameter $b_R$ was

significantly higher compared to $b_P$ ($\Delta b$ = -0.16 ± 0.13, $t_{59}$ = -9.61, $p < .001$, Cohen's $d$ = -1.24), whereas temperatures $\tau_P$ and $\tau_R$ did not differ significantly ($\Delta\tau$ = -0.01 ± 0.06, $t_{59}$ = -1.65, $p = .104$, Cohen's $d$ = -0.21). Thus, even in the full model, there is no evidence for a change in decision temperature.

To check the sensibility of the fitting procedure, we performed parameter recovery on simulated data sets generated using the fitted parameters from the best-fitting bias model and tested whether we could recover the ground-truth parameters based on these simulated data. Results showed successful recovery of all model parameters, as evidenced by the high correlations between fitted and recovered parameters (see section *Parameter recovery* and Section 3.2. in S1 Appendix). Next, we validated the winning model. Due to the confound in the SDT-based analyses of the behavioural data, any model which includes learning generates a difference between pre-probe and probe trials, hence, examining qualitative difference between reinforced and probe data for model validation is not warranted. Thus, we used the simulated go-response probabilities for go and no-go trials to validate which parameters are necessary to recapitulate the patterns observed in participants' behaviour (see Figs J and K in S1 Appendix). We found that only the winning model that included separate bias parameters for blocks with and without reinforcement could replicate the observed difference in go-response probabilities between reinforced and probe trials (see Fig 3E and Fig L in S1 Appendix). Taken together, parameter recovery and model validation indicate that our model with two different bias parameters and one fixed softmax temperature provided a plausible account of participants' behaviour in our experiment.

The increase in the sensitivity index *d'* from reinforced to probe blocks, resulted from a differential reduction in hit versus false alarm rates. It may appear surprising that a mere change in response bias is sufficient to drive such differential changes. However, this arises naturally as a consequence of the sigmoid shape of the softmax choice function, together with a positive initialization of value estimates. This effect is shown in Fig 3D depicting the softmax choice functions for the reinforced and probe blocks (based on the fitted values for $\tau$, $b_P$ and $b_R$). For the vast majority (50 of 60) of our participants, we found positive estimates for initial values ($Q_0$ = 0.37 ± 0.58, $t_{59}$ = 4.94, $p < .001$, Cohen's $d$ = 0.64, Fig 3C), reflecting participants' tendency to act (i.e., providing a go-response) on the first trials of the experiment. Thus, the initial value estimate is already shifted from zero (dashed line) to higher values (grey line). The difference between the two curves describes the reduced go-response probability for probe trials compared to reinforced trials. This difference is smaller for go stimuli than for no-go stimuli; During the acquisition of values for go stimuli, both curves quickly converge towards 1 (green arrow), while the exact opposite happens for no-go stimuli. These likewise start at a relatively high positive value, but because they are updated in the opposite direction during learning, the difference between the two curves first increases before decreasing again when converging towards -1. Thus, a positive value initialization together with a decrease in action bias results in enhanced instrumental performance during probe blocks, without any change in choice sensitivity to acquired value.

## Discussion

When evaluating learning success, instrumental performance is measured during the learning process, conflating measures of learning with proficiency in expressing the acquired knowledge. However, it is well known that learning of action-outcome association relies, in part, on different neural substrates than expression of instrumental performance contingent upon these associations. Specifically, some neural mechanisms required for learning are not involved in the expression of learnt behaviour and vice versa [6,7,8,24,25,26,27]. To disentangle these

two concepts, the phases designed to assess learning versus expression of task performance are usually separated by a considerable delay in these studies. This implies that behaviour during the test phase relies on long-term consolidation of memories. Alternatively, to obtain a pure measure of an agents' current learning success, one option is to omit reinforcement/feedback during the learning process, which yielded inconsistent results in previous studies [10,11,12,28].

We aimed to reconcile the apparent contradictory results by investigating whether removing reinforcement unmasks latent associative knowledge during instrumental learning. Healthy humans performed an instrumental go/no-go learning task with reinforcement in which blocks without reinforcement were interspersed. Replicating previous rodent work [12], we first found that the sensitivity index *d'* was enhanced during blocks in which reinforcement was omitted. However, these findings based on signal detection theory are confounded with trial number, as measures like *d'* are computed over windows of several trials. This is problematic for dynamic processes like learning, where earlier sets of trials are inherently disadvantaged compared to later sets, unless learning has reached a plateau. To avoid this confound, we therefore used reinforcement learning models to investigate the mechanism driving the apparent change in choice sensitivity during non-reinforced trials. Such models have the advantage that they provide a point estimate for the stimulus value on each trial. Contrary to our expectations, better performance in non-reinforced trials did not result from an increased choice sensitivity, as would be reflected in a decreased softmax choice temperature. Instead, our modelling results suggest that the change in *d'* can fully be accounted for by a decrease in a bias parameter (reflecting participants' overall propensity to act), together with a positive value initialization (reflecting participants' tendency to act on the first trials). First, a model with only a single softmax temperature but with two separate bias parameters (for reinforced and non-reinforced blocks, respectively) provided the best fit to participants' choices. Second, this model successfully recapitulated the behavioural patterns we observed. Third, even when we explored a full model (with two separate bias and temperature parameters), we found that, while bias parameters still differed significantly between non-reinforced and reinforced blocks, temperatures did not differ. Altogether our results suggest that the omission of feedback led to an adjustment of response strategy rather than enhancing the expression of latent task knowledge.

The effects of removing feedback may be dependent on the domain of learning being studied. In the domain of perceptual learning, one study reported impaired performance without feedback [10], whereas others reported no performance differences between reinforced and non-reinforced trials in a similar paradigm [11,28]. In the domain of associative learning, one recent rodent study suggests a beneficial effect of omitting reinforcement on performance in an instrumental learning task [12]. We based our learning paradigm on this latter work and also found enhanced performance during non-reinforced trials (measured using the sensitivity index *d'*) in our sample of healthy volunteers. Therefore, despite interpreting the findings differently, we conceptually replicated the rodent work in humans.

Due to challenges inherent to translational work, one might argue that differences in task environments between the rodent tasks used by Kuchibhotla and colleagues' [12] and our human task could lead to results relying on different mechanisms. For example, extinction is well known to occur when animals' actions are not reinforced (as in the probe trials) [29,30]. To prevent extinction effects to confound results, only short probe phases were used in the rodent study [12], making it unlikely that their effects are influenced by extinction. Because rodents did not know that their knowledge is tested in probe trials, we aimed to adapt instructions accordingly. To avoid that participants try to perform better in probe trials, we instructed them to continue responding to the task in non-reinforced trials as they did in reinforced trials.

Finally, these findings hinge on the specific task structure. Our task encouraged an asymmetric response pattern: Performing an action (i.e. go response) is the only possibility to learn and explore possible outcomes for specific stimuli in our task, as refraining from responding yielded no feedback. Therefore, go responses provide a gain in information (information bonus), making them advantageous relative to no-go responses. This information bonus may account for the two main effects our computational model revealed: On the one hand, the information bonus is greatest early in the experiment, because values are not reliable learnt yet. This is reflected in the positive value initialization in our computational model. On the other hand, information can only be obtained in reinforced trials, resulting in a more pronounced general bias to withholding responses in non-reinforced trials. The resulting behaviour is in line with research about curiosity, suggesting that a lack of information makes individuals curious and facilitates information seeking behaviour [31]. Different tasks that do not encourage go responses for optimal performance are unlikely to result in a beneficial effect of reinforcement removal. Specifically, environments that do not favour any specific response (e.g., situations where the outcomes of both choice options are always presented) will result in a neutral value initialization, because participants do not need to perform a go response in order to gain information. Any change in response bias (if present) would affect hit and false alarm rates to the same extent, thus resulting in no difference in performance. Following the same logic in reverse, environments favouring no-go responses should result in a negative value initialization, thus resulting in the opposite pattern: a reduced performance during non-reinforced trials.

Likewise, we assume that an adaptive response bias would also manifest in foraging-related tasks. In previous work, we have shown that the average rate of responding is dependent on the local average reward rate in the environment, even when rewards were not contingent on participants' choices [32]. Furthermore, similar effects may be expected for learning environments characterized by high levels of volatility, where contingencies between choices and outcomes change frequently. In such situations, an agent would benefit from a high response probability on reinforced trials. Conversely, the cost of a false alarm would be greater than the cost of missing an opportunity in non-reinforced trials, resulting in decreased go-response probabilities. Thus, at high levels of environmental volatility, the effect of feedback removal on the response bias might be even more pronounced, as frequent contingency changes further increase the informative value of go-responses during reinforced periods. Therefore, we suggest that the effects of feedback removal on instrumental performance are highly dependent on the particular characteristics of the task at hand.

In conclusion, we found that omitting feedback during learning may indeed improve instrumental performance. However, our results show that this improvement results from a shift in participants' overall bias to act, rather than from unmasking of task knowledge.

## Materials and methods

### Ethics statement

All studies were approved by the Ethics Committee for Noninvasive Research on Humans of the Heinrich Heine University Düsseldorf (reference OR01-2020-01).

### Participants

Participants were recruited from the local student community of the Heinrich Heine University, Germany. Each experiment was run with healthy participants with normal or corrected-to-normal vision and no history of neurological diseases. Participants signed a written informed consent prior to participation and received course credit or monetary compensation

for their participation. Participants were only included if they succeeded at learning during the task. For that reason, we set a performance criterion of $d' \geq 1$, which participants needed to exceed across all reinforced trials of the second half of the experiment. We a priori defined two criteria to constrain the sample size for each experiment: (1) collecting data of at least 30 participants fulfilling the performance criterion, and (2) using a Bayesian stopping rule, meaning that we set out to acquire as many participants as needed to find strong evidence either against or in favour of an overall change in $d'$ between reinforced and probe trials ($BF_{01} > 10$ or $BF_{10}$ > 10, respectively). For Experiment 1, 46 participants volunteered to take part in the study and 16 were excluded as their behaviour did not fulfil the predefined performance criterion. In Experiment 2, 39 participants completed the task, 9 of which did not pass the criterion and were therefore excluded. For both experiments, we found strong evidence in favour of an overall change in $d'$ between reinforced and probe trials after inclusion of 30 participants, such that data acquisition in both experiments was stopped after 30 included participants. This resulted in $N = 30$ participants (mean age = 21.2 ± 4.0, 22 female and 8 male) for Experiment 1 and $N = 30$ participants (mean age = 24.5 ± 5.0, 9 female and 21 male) for Experiment 2. As both experiments yielded qualitatively similar results, we pooled the data from both studies for further analyses. Results are reported separately for the two studies in the supplementary materials (see Section 1 in S1 Appendix).

## Behavioural task

As paradigm we employed a visual go/no-go learning task written and presented with PsychoPy (version 3.1.5) [33]. Stimuli were presented on an Asus PG248Q LCD display (24", 1920x1080, 60 Hz refresh rate) at a viewing distance of 80 cm. Each trial started with the presentation of a fixation cross, spanning 0.72˚ visual angle, for a pseudo-randomly selected duration of 1000 ms to 1600 ms to keep the participants' attention to the centre of the screen. Stimuli spanned the central 3.58˚ visual angle of the screen and were presented for 500 ms. We used greebles (Greebles 2.0) [13] as stimuli. Greebles are three-dimensional objects which are usually used for object and face recognition. Based on the body shape, greebles are classified into so-called families, while three other features vary for each exemplar, such that similarity between exemplars of the same families is larger than between exemplars across families. To drive slow learning, we used twelve greebles that were evenly sampled from three families. For each participant, half of the greebles of each family were pseudo-randomly assigned to be go stimuli while the other half were no-go stimuli to make sure that learning is similarly difficult across participants. They had to learn to perform a button press for go stimuli and to withhold a response for no-go stimuli. The response could be administered during stimulus presentation and within 1000 ms after stimulus offset. The duration of this response window was well sufficient to perform a go-response (see Fig M in S1 Appendix). Feedback was immediately presented after button press consisting of a smiley or frowny (spanning 1.93˚ visual angle each) for correct and incorrect actions, respectively. Every correct button press was rewarded with 2 cents, which was indicated by an increase of the progress bar, while every incorrect button press led to a 2 cents deduction and a decrease of the progress, respectively. To design the task comparable to previous animal experiments, no feedback was delivered (and neither money won nor lost), when no button was pressed. The task consisted of 588 trials, with each of the twelve stimuli presented 49 times in pseudo-randomized order with the constraints that each one is presented once in twelve trials and that two consecutive stimuli were always different. Reinforced trials were interspersed by five probe blocks (36 trials each). Participants were instructed to respond as they had previously done during the reinforced blocks, but they no longer received feedback for their choices and the progress bar disappeared. The paradigm

was performed in five phases of 120 trials. Participants were encouraged to rest between the phases as long as they needed.

## Behavioural analyses

Data analyses were conducted using MATLAB (MATLAB Version R2016b, Massachusetts: The Mathworks Inc.). Statistical significance testing, including the computation of the Bayes Factor of the main effect, was done in JASP (JASP Team (2019). JASP (Version 0.11.1) [macOS]). We calculated the sensitivity index $d'$ as:

$$d\prime = z(HR) - z(FAR) \qquad [1]$$

With the $z$-scored hit rate $z(HR)$ and the $z$-scored false alarm rate $z(FAR)$ [16]. As ceiling performances cannot be $z$-scored, we used $HR = 1 - \frac{1}{2N}$ as correction for $HR = 1$, and $HR = \frac{1}{2N}$ as correction for $HR = 0$, with the number of trials $N$. Because we calculated $d'$ for a small number of trials, the use of corresponding trials for correction may result in underestimated $d'$ values [34], so we used a correction allowing $HR$ and $FAR$ to be approximately 0 or 1. The negative bias criterion $c$ was calculated as:

$$c = \frac{1}{2}\left(z(HR) + z(FAR)\right) \qquad [2]$$

For comparison of $d'$ between probe and reinforced trials, we defined the 36 trials (the length of a probe block) before reinforcement removal as pre-probe trials, and the 36 trials after reinforcement was reinstated as post-probe trials, and computed the difference between probe and pre-probe trials, and between probe and post-probe trials. When visualising the learning curve, $d'$ was computed within a sliding window of 21 trials. To further specify the effects of $d'$ and the bias criterion, we also analysed hit and false alarm rates separately for reinforced and probe trials.

Changes in behaviour between reinforced and probe blocks were analysed using one-sample Student's $t$-test comparing their difference against zero. The $t$-tests examined the null hypothesis that there is no difference in behaviour between reinforced and probe blocks at a significance level of $\alpha = 0.05$. To test whether the change in behaviour from reinforced to probe blocks differed across successive probe blocks, we subjected the differences in $d'$ (probe —pre-probe, and analogously for hit and false alarm rates) to repeated measures ANOVAs (Greenhouse-Geisser correction to adjust for lack of sphericity when $\varepsilon < 1$).

Go-response probabilities $P(Go)$ were computed with a sliding window of five trials and averaged over participants for visualization. We used a smaller window size compared to the $d'$ learning curves, because it was sufficient to obtain a good resolution for go-probability.

## Reinforcement learning models

Computational modelling was performed in MATLAB (MATLAB Version R2016b, Massachusetts: The Mathworks Inc.). Altogether, we tested four models, in each of which values for chosen stimuli were updated using a delta update rule:

$$Q_{i,t+1} = Q_{i,t} + \alpha(r_t - Q_{i,t}) \qquad [3]$$

Where $Q_{i,t}$ is the value for stimulus $i$ presented on trial $t$, $\alpha$ is the learning rate and $r_t$ is the observed outcome on trial $t$. On trials during reinforced blocks in which subjects performed a go-response, $r_t$ was either -1 or 1, depending on whether the chosen stimulus was a go- or no-go stimulus, respectively. When participants gave a response during probe trials, the value of

the corresponding stimulus was not updated, i.e. $Q_{i,t+1} = Q_{i,t}$. In line with the non-monotonic plasticity hypothesis [22], we have recently shown that associations of unchosen stimuli are weakened [23]. Therefore, on trials during which participants performed a no-go response, we assumed passive forgetting of the displayed option governed by a decay parameter $\theta$ (for reinforced and probe trials):

$$Q_{i,t+1} = \theta Q_{i,t} \qquad [4]$$

Initial $Q$-values $Q_0$ for the first trial were also treated as a free parameter. The learning rate $\alpha$ was not treated as a free parameter and instead fixed at $\alpha = 0.06$ for all participants (see section *Fixed learning rate* for explanation). Choices were modelled using a softmax choice rule. The four models differed with regard to the bias and temperature terms contained in their respective softmax choice rules:

**Baseline model.** Choices in the baseline model are generated using a softmax choice rule that contains a single temperature $\tau$ and bias term $b$:

$$p_t = \frac{1}{1 + e^{\frac{-(Q_{i,t}+b)}{\tau}}} \qquad [5]$$

Where $b$ is a general bias to act and $\tau$ the temperature parameter determining the stochasticity of action selection. Importantly, the softmax choice rule used the same parameters for reinforced and probe trials, thus, the baseline model did not discriminate between block types. Altogether, for this model, four parameters were thus fit: the initial $Q$-value $Q_0$, the decay $\theta$, the general bias $b$ and the softmax temperature $\tau$.

**Temperature model.** Like the baseline model, the temperature model also contained a single bias parameter $b$, but it allowed for the temperature $\tau$ to vary between reinforced and probe blocks:

$$p_t = \frac{1}{1 + e^{\frac{-(Q_{i,t}+b)}{\tau_k}}} \qquad [6]$$

Where $\tau_k = \tau_R$ for reinforced trials and $\tau_k = \tau_P$ for probe trials. Thus, for this model, five parameters were fit: the initial $Q$-value $Q_0$, the decay $\theta$, the general bias $b$, the softmax temperature $\tau_R$ for reinforced trials and the softmax temperature $\tau_P$ for probe trials.

**Bias model.** Here, instead of allowing the temperature to vary between reinforced and probe trials, we now allowed for separate bias parameters:

$$p_t = \frac{1}{1 + e^{\frac{-(Q_{i,t}+b_k)}{\tau}}} \qquad [7]$$

Where $b_k = b_R$ for reinforced trials and $b_k = b_P$ for probe trials. Thus, for this model, five parameters were fit: the initial $Q$-value $Q_0$, the decay $\theta$, the general bias $b_R$ for reinforced trials, the general bias $b_P$ for probe trials and the softmax temperature $\tau$.

**Full model.** This model is a combination of the temperature and bias model: It used both a separate temperature and a separate bias for reinforced and probe trials:

$$p_t = \frac{1}{1 + e^{\frac{-(Q_{i,t}+b_k)}{\tau_k}}} \qquad [8]$$

Thus, for this model six parameters were fit: the initial $Q$-value $Q_0$, the decay $\theta$, the general bias $b_R$ for reinforced trials, the general bias $b_P$ for probe trials, the softmax temperature $\tau_R$ for reinforced trials and the softmax temperature $\tau_P$ for probe trials.

**Model comparison.** Models were fit by minimizing the negative log likelihood *NLL*:

$$NLL = -\sum_t log\ (p_t) \tag{9}$$

Where $p_t$ is a vector containing, for each trial $t$, the model's probability to select the choice performed by the participant. We used unconstrained non-linear optimization implemented in Matlab's function *fmincon*. To minimize the risk of finding local optima, we started optimization from 1000 random starting points for each participant. To account for the different number of parameters, we used the Bayesian information criterion for model comparison:

$$BIC = 2NLL + n_{param} log(n_t) \tag{10}$$

Where $n_{param}$ and $n_t$ are the number of parameters and trials, respectively. A lower *BIC* score indicates a better model fit.

## Model validation

To test whether the best-fitting model provides a good account of participants' behaviour, we tested whether we could replicate the behavioural results using simulated datasets. Because the comparison of pre-probe and probe trials confounds performance and trial number, the replication of the SDT-based behavioural analysis using simulated data is not suited for model validation. Instead, we visually inspected simulated data for the baseline model, the temperature model and the bias model. To this end, we simulated 500 datasets per participant based on these models, using the parameter combination fitted for the respective participant. Then, we computed go-response probabilities for go and no-go trials to investigate how single parameters of the different models changed these probabilities and ultimately, if a bias parameter for both reinforced and probe blocks is necessary to describe the patterns found in behaviour (see Section 3.3. in S1 Appendix). For visualization, go-response probabilities were averaged over the number of simulations and simulated participants.

## Parameter recovery

In order to test the reliability of fitted parameters, we performed a parameter recovery. We used the 500 simulated datasets per participant from the model validation and fitted the bias model in the same way as described above for the experimental data. For each participant and each parameter, we compared the original model fit with the synthetic model fit (see Fig I in S1 Appendix). The high correlation coefficients (all $\rho > 0.99$, see Table L in S1 Appendix) indicated successful recovery for all parameters.

## Fixed learning rate

The deterministic task design gives rise to a strong anti-correlation between learning rate and softmax temperature, thus, both parameters could not be estimated by the models independently [21]. Note however, that it is not plausible to assume different learning rates for probe and reinforced blocks, as no learning rate is applied to the probe trials. Instead, it was our goal to test whether the sensitivity of choices to acquired values changes between reinforced and probe trials. Therefore, we fit the models using a fixed learning rate of *α = 0.06* for all participants. In order to ascertain that the results are independent of this particular choice of learning rate, we used six different learning rates evenly log-spaced between 0.01 and 0.20. With this set of learning rates, we fitted all four models and compared them. We found that the bias model performed best, while the temperature model was the worst fitting model, independent of the learning rate (see Section 3.1. in S1 Appendix).

## Supporting information

**S1 Appendix. Further Analyses of behavioural and computational data.** In Section 1, we present results separately for the two experiments. In Section 2, we show further analyses of the pooled dataset. In Section 3, analyses of the computational modelling, in particular the control analysis for the fixed learning rate, the parameter recovery and the model validation, are presented. In Section 4, we analysed reaction times.
(PDF)

## Acknowledgments

We thank Judith Geusen, Christina Kalinicenko, Joshua Saal and Georg Schäfer for their support during data acquisition. Computational infrastructure and support were provided by the Centre for Information and Media Technology at Heinrich Heine University Düsseldorf.

## Author Contributions

**Conceptualization:** Hannah Kurtenbach, Eduard Ort, Monja Isabel Froböse, Gerhard Jocham.

**Data curation:** Hannah Kurtenbach, Eduard Ort.

**Formal analysis:** Hannah Kurtenbach, Eduard Ort, Monja Isabel Froböse, Gerhard Jocham.

**Funding acquisition:** Gerhard Jocham.

**Investigation:** Hannah Kurtenbach.

**Methodology:** Hannah Kurtenbach, Eduard Ort, Monja Isabel Froböse, Gerhard Jocham.

**Project administration:** Hannah Kurtenbach, Eduard Ort, Monja Isabel Froböse, Gerhard Jocham.

**Software:** Hannah Kurtenbach, Eduard Ort, Gerhard Jocham.

**Supervision:** Eduard Ort, Monja Isabel Froböse, Gerhard Jocham.

**Validation:** Hannah Kurtenbach, Eduard Ort, Monja Isabel Froböse, Gerhard Jocham.

**Visualization:** Hannah Kurtenbach.

**Writing – original draft:** Hannah Kurtenbach.

**Writing – review & editing:** Eduard Ort, Monja Isabel Froböse, Gerhard Jocham.

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
