## [Decision Letter · Decision Letter 0]

6 Jul 2022

Dear Dr  Kurtenbach,

Thank you very much for submitting your manuscript "Removal of reinforcement improves instrumental performance in humans by decreasing a general action bias rather than unmasking learnt associations" for consideration at PLOS Computational Biology.

As with all papers reviewed by the journal, your manuscript was reviewed by members of the editorial board and by several independent reviewers. In light of the reviews (below this email), we would like to invite the resubmission of a significantly-revised version that takes into account the reviewers' comments.

We ask to take into great account all the reviewers' remarks (major revision - the paper will be send back to the original reviewers), but we also wanted to mention to put extra-care in justifying the choices of the behavioural metrics (while presenting also more standard metrics of performance), the modelling assumptions and and exclusion criteria.

We cannot make any decision about publication until we have seen the revised manuscript and your response to the reviewers' comments. Your revised manuscript is also likely to be sent to reviewers for further evaluation.

Sincerely,

Stefano Palminteri

Associate Editor

PLOS Computational Biology

Samuel Gershman

Deputy Editor

PLOS Computational Biology

Reviewer's Responses to Questions

**Comments to the Authors:**

Reviewer #1: This work investigates the potential performance benefits of removing reinforcement during learning. To do so, the authors subjected 60 healthy volunteers to a novel visual go/nogo learning task with deterministic contingencies, and with blocks with and without feedback (i.e. no reinforcement). Performance was quantified by the sensitivity index d’.

As expected, blocks with no-reinforcement showed improved instrumental performance. However, this improvement was not due to increased sensitivity to learned value (i.e. unmasking task knowledge) but to a reduction in overall propensity to act (i.e. reduced response bias).

This is a very interesting study, which reconciles seemingly contradictory results in humans (omitting feedback impairs learning performance) and animals (omitting feedback improves learning performance). The paper is very clearly written, the hypotheses are very clearly stated, and the methods seem sound – a combination of behavioural experiments and state-of-the-art computational models and modelling procedures (model comparison, parameter recovery, simulation). The findings appear to be supported by the data. My only concern is the interpretation of the results and their generalization to other learning environments, where exploration and risk-taking, for example, may be beneficial.

1. Here, the results suggest that the better performance observed on trials without feedback can be explained by a reduced propensity to act. I wonder to what extent this result is task-specific. Adopting cautious response strategies (that minimize false alarms) may be beneficial in some learning environments, but detrimental in others. For example, in richer or less risky environments, a high propensity to act may be beneficial because the cost of missing an opportunity is greater than the cost of a false alarm. Would you expect the same results with different reward probabilities/magnitudes?

2. Related to the previous comment: young children sometimes engage in "motor babbling”, a self-exploring behaviour that is often described as an excessive propensity for (random) action. This propensity to act has a learning function: it allows the *accidental* discovery of action-effect relationships, which can then be implemented by the child *intentionally*. So I wonder if the benefit of a reduced response bias observed in your task is not due to the fact that the task does not particularly reward this type of self-exploring behavior. Exploration can also be useful in environments with frequent and unpredictable contingency reversals. Would you expect your findings to hold even in environments where exploration is highly advantageous?

3. And more generally: couldn't the apparently contradictory results observed between humans and rodents (improved vs impaired performance when omitting feedback) be due to differences in task environments?

Reviewer #2: This is an interesting study using a behavioural task in healthy human volunteers and computational modelling. The authors investigate an effect known from rodent studies in which accuracy on an instrumental task increases in blocks in which no feedback is given, compared to blocks in which animals are both making choices and learning the value of items in parallel. The authors replicate this effect in human observers using a go/no-go task. Through careful use of modelling the authors show that improved performance in no-feedback (‘probe’) blocks is best explained in terms of a reduction in response-willingness, or response bias as they call it. This impacts on both types of active response (hits and false alarms) but it affects false alarms more, because participants have, especially in early blocks of the task, a positive response bias overall (they respond more often than they optimally should).

I thought this was interesting, carefully done and clearly presented. I find no errors per se. My main comments would be about the interpretation – I would like to invite the authors to think about these and comment in the manuscript text, if they find it appropriate to do so.

Firstly, I wonder if humans and rodents treat the no-feedback blocks in the same way. In humans, we call them probe blocks as we are testing the observers knowledge, and the observer knows this. Thus we could expect the observers to become more cautious about making errors and try harder in the probe blocks. In contrast, rodents do not understand this demand characteristic – we sometime call no-feedback blocks in rodent tasks testing in extinction’ and expect their knowledge of the stimulus-outcome contingencies to decay in this time. Given this conceptual difference how sure can we be that the human task taps the same constructs as even the exact same task done in rodents?

Secondly, I think the effects could be explained in terms of an information bonus associated with active responding (if participants always withhold responding they will never learn which greebles are rewarded) – which is greatest early in the experiment, and absent on probe trials. Maybe the reason for the positive response bias is the information bonus. I think this possibility should at least be addressed in the text (even if the authors have aa counter-argument). If this is the explanation it doesn’t diminish the value of the current study but does suggest follow-ups (what would happen if you always showed the counterfactual on non-response trials)?

Line 133

“We compared performance of the 36 trials in a probe block with performance in the 36 trials before each probe block (pre-probe trials).”

Isn’t this confounded with trial number though? There is a general rising trend in d’ so always comparing probe with pre-probe (rather than post probe) should lead to probe having a higher d’. I realise people can’t be learning during the probe trials but they will he learning during the preceding 30 trials, so the average of 30 pre-probe trials underestimates the instantaneous d’ at the start of the probe block.

That said I see d’ is lower in the post probe trials than the probe trials so I do believe there is a real increase in performance in the probe blocks.

I see later this confound is pointed out and (hopefully) resolved by the introduction of the model. Nevertheless I find it odd to read the first section (pre-model) and not see this confound acknowledged or dealt with. Maybe some rephrasing, or an acknowledgement of the issue and an indication that the model will later address this issue is in order?

Line 214

“On probe trials in which the go action was selected (and no feedback received), no changes were applied to value estimates. During trials in both reinforced and probe blocks without a go action, we assumed that values were subject to passive forgetting (21,22) via diffusion towards zero governed by the decay parameter θ.”

Why are no-feedback go trials not subject to decay?

Line 234 f3c

Initial value estimates are optimistic. Conceptually, could this reflect an information bonus rather than an optimistic value estimate per se? Optimistic value estimate means (I think) that participants respond ‘go’ more often than they should. But they only find out the value of the greeble (whether positive or negative) if they respond – withholding the response = no information.

Furthermore, the information bonus should be absent in probe trials, perhaps this could drive the reduced (overall) tendency to respond in probe trials

Reviewer #3: In this study, the authors investigate human reinforcement learning using a Go/No-Go deterministic task performed by a total of 60 participants over 2 experiments, in an attempt to replicate previous results recently found in rodents (Kuchibhotla et al. Nat Commun 2019). The results from the original study were partly replicated in humans: when comparing blocks without feedback (‘probe’ trials) to blocks where feedback was provided (‘reinforced’ trials), the measured sensitivity increased. This was due to a decrease of both the false alarm rate (button press in No-Go trials) and the hit rate (button press in Go trials), the latter effect not having been found in rodents. Next, the authors turn to computational modeling and elegantly show that the behavioral effects result from a shift in the action bias in probe trials, rather than an increased sensitivity to the learnt option values. Overall, the paper is clear and well written. This work represents an important and relevant research topic in the field, however I have some concerns regarding the behavioral interpretations and modeling procedures that I think the authors should address.

Behavioral results

1) One major concern pertains to the definition of performance. In this task, a button press is rewarded in Go trials, while it is punished in No-Go trials. Therefore, an accurate measure of the performance would be the proportion of button presses in Go trials and withholding in No-Go trials, yet in the paper the ‘performance’ is approximated by the sensitivity d’, which is the difference between normalized hit rate (HR) and false alarm rate (FAR). In the original study, the increased sensitivity relied on no change in the HR but a decrease in FAR (hence, increased performance). In this study however, both HR and FAR decrease, which means that the performance increases in No-Go trials BUT decreases in Go trials. At best, this should be addressed and discussed in the manuscript; at least, the manuscript will benefit from displaying the actual performance changes between probe and reinforced trial blocks.

2) One of the key results from the original study was that not only does the FAR decreased during probe blocks compared to pre-probe blocks, but it also increased again in post-probe blocks. This result is partly replicated here, since it was only the case in one of two experiments. I believe this should be mentioned in the manuscript (lines 141, 189-190), especially since the authors chose to present pooled results for both experiments (even though some results do not replicate). Yet, I believe that presenting the results of both experiments separately in the supplementary materials is good practice and should be encouraged.

Computational results

3) My main remark concerning the modeling part of the manuscript, is that the results seem to come as a surprise, while they could have easily been expected from the behavioral results. This partly comes from the definition of the sensitivity as being a measure of performance (see point 1): based on the model description, increasing the temperature would make the choice more stochastic, i.e., affect the behavior in Go and No-Go trials in opposite directions (decrease p(hit) in Go trials, but increase p(false alarm) in No-Go trials). This would indeed affect the performance but not the sensitivity. To modify the behavior towards more withholding, the simplest way is to add a shift in the decision function (not to modify stochasticity), i.e., a bias towards not responding, as evidenced by model fitting.

4) Fig 3: related to the previous point, this might change with the Q-value initialization. It seems a bit confusing to have a positive Q0 (i.e., a bias towards acting) AND a negative bias parameter (i.e., a bias towards withholding). Have the authors looked into the possible correlations between parameters? Or parameter values when Q0 is fixed at 0 (i.e., no initial bias)?

5) Line 269: based on Fig 1, I wouldn’t say that the model accurately captures the behavioral patterns: the plateau seems to be achieved much sooner. One explanation could be the fixed value which was chosen for the learning rate: although it makes sense to fix that value in the context of this task, it is not clear from the methods, nor from the results, why the value of 0.06 was chosen (since almost all values from tables S11 and S12 seem to produce similar behavioral patterns). Relatedly, the manuscript could benefit from simulations from other models, in order to grasp why the parameters of the winning were needed to produce such behavioral patterns (for reference, see Palminteri et al. TICS 2017). For example, while I agree with the statement on lines 285-287, it will be useful to see that the other proposed models don’t achieve this (if, indeed, they don’t).

6) Line 506: it is not clear from the methods whether this passive forgetting is performed on the Q-value of the displayed option, or all others as well. In which case, (1) this could be extended to all trials (including Go trials) and (2) the weakening of value associations can also be achieved with a forgetting rule such as those from Collins and Frank, Eur J Neuro 2012, where the unseen options decay towards their initial value (if the initial Q-value differs from 0, why would forgetting diffuse towards zero?).

Methods

7) I have some concerns about the inclusion/exclusion criteria. Not only are the criteria defined on the one dependent variable of interest (i.e., the ‘performance’, which, again, is a measure of sensitivity and not formally task performance), but it led to the exclusion of 34% and 23% of participants in each experiment, which is a very high relative number of rejections. How do the results of the whole dataset look like? Why did the authors choose to stop at 30 participants per experiment? Did the sample of 30 participants reach the BF threshold? In which direction?

8) In this task, the stimuli were presented for 500ms, and participants had 1000ms to either press a button or withhold from doing so. The time window is shorter than those used in similar tasks (e.g., Guitart-Masip et al. Neuroimage 2012, and following studies, where the total reaches 2500ms). Could a different time window have influenced the results (e.g., no action bias towards button press)?

Minor points

9) Line 199: I suggest not to mention ‘value differences’ in this case, since the decision is based on the value of one option only.

10) Line 202: in general, the temperature should not be perceived as a measure of performance, since in multi-alternative decisions, it rather balances between exploration and exploitation based on the knowledge at trial t (i.e., if the values are not accurately represented, a lower temperature might lead to more errors).

11) Line 315: the use of the term ‘optimistic’ might be confusing here, since it is also used in other reinforcement learning tasks as the difference in learning rates from positive or negative prediction errors, not the initial option value or bias towards acting.

12) Line 364: post-learning phases do not necessarily happen on a separate day in human reinforcement learning (see e.g., Frank et al. Science 2004 and following studies; Palminteri et al. Nat. Commun. 2015 and following studies).

13) Line 403-405: this is a surprising assumption. Based on behavioral and modeling results, the performance, i.e., withholding from button press, should increase in No-Go trials (regardless of whether there are other types of trials). It is also unclear why the first trial option value should be different from the one in a task featuring both trial types.

14) Line 473: the correction for extreme HR is usually based on the number of corresponding trials only (see Macmillan and Creelman 2004, and the original publication), although this might be of little relevance here.

**Have the authors made all data and (if applicable) computational code underlying the findings in their manuscript fully available?**

Reviewer #1: None

Reviewer #2: Yes

Reviewer #3: **No: **All behavioural data will be made publicly available on OSF upon publication.

PLOS authors have the option to publish the peer review history of their article (what does this mean?). If published, this will include your full peer review and any attached files.

Reviewer #1: No

Reviewer #2: No

Reviewer #3: No
---

## [Decision Letter · Decision Letter 1]

24 Nov 2022

Dear Dr Kurtenbach,

We are pleased to inform you that your manuscript 'Removal of reinforcement improves instrumental performance in humans by decreasing a general action bias rather than unmasking learnt associations' has been provisionally accepted for publication in PLOS Computational Biology.

Best regards,

Stefano Palminteri

Academic Editor

PLOS Computational Biology

Samuel Gershman

Section Editor

PLOS Computational Biology

Reviewer's Responses to Questions

**Comments to the Authors:**

Reviewer #1: I think the authors did an excellent job responding to the reviews. I congratulate them on this important contribution to literature.

Reviewer #3: The authors have done an excellent job addressing my comments in a very constructive manner. The additional analyses are convincing and I am happy with the revised manuscript. Regarding point 7 on excluded participants, I do not think it is essential to include the information in the supplements.

**Have the authors made all data and (if applicable) computational code underlying the findings in their manuscript fully available?**

Reviewer #1: Yes

Reviewer #3: None

PLOS authors have the option to publish the peer review history of their article (what does this mean?). If published, this will include your full peer review and any attached files.

Reviewer #1: No

Reviewer #3: No

---

## [Editor Report · Acceptance letter]

3 Dec 2022

PCOMPBIOL-D-22-00740R1 

Removal of reinforcement improves instrumental performance in humans by decreasing a general action bias rather than unmasking learnt associations

Dear Dr Kurtenbach,

I am pleased to inform you that your manuscript has been formally accepted for publication in PLOS Computational Biology. Your manuscript is now with our production department and you will be notified of the publication date in due course.

With kind regards,

Zsofia Freund
